
# Processes and controls of regional floods over eastern China

Yixin Yang[1,2], Long Yang[1,2]*, Jinghan Zhang[1,2], Qiang Wang[1,2]

[1] School of Geography and Ocean Science, Nanjing University, Nanjing, China
[2] Frontiers Science Center for Critical Earth Material Cycling, Nanjing University, Nanjing, China

*Correspondence to*: Long Yang (yanglong@nju.edu.cn)

**Abstract.** Mounting evidence points to elevated regional flood hazards under a changing climate, but existing knowledge about their processes and controls is albeit limited. This is partially attributed to inadequate characterizations of the spatial extent and potential drivers of these floods. Here we develop a machine-learning based framework (mainly including the density-based clustering algorithm DBSCAN and conditional random forest model) to examine the processes and controls of

regional floods over eastern China. Our empirical analyses are based on a dense network of stream gauging stations with continuous observations of annual maximum flood peak (i.e., magnitude and timing) during the period 1980-2017. A comprehensive catalog of 318 regional floods is developed. We reveal a pronounced clustering of regional floods in both space and time over eastern China. This is dictated by cyclonic precipitating systems and/or their interactions with topography. We highlight contrasting behaviors of regional floods, in terms of their spatial extents and intensities. These contrasts are

determined by fine-scale structures of flood-producing storms and anomalous soil moisture. While land surface properties might play a role in basin-scale flood processes, it is more critical to capture spatial-temporal rainfall variabilities and soil moisture anomalies for reliable large-scale flood hazard modeling and impact assessments. Our analyses contribute to flood science by better characterizing the spatial dimension of flood hazards and can serve as basis for collaborative flood risk management under a changing climate.

**1 Introduction**

Riverine floods evolve in both space and time (Blöschl, 2022). Floods that occur simultaneously over a collection of neighboring basins are interchangeably termed as widespread floods (e.g., Brunner et al., 2020a), trans-basin floods (Uhlemann et al., 2010), multi-basin floods (De Luca et al., 2017), or synchronous floods (Berghuijs et al., 2019). Here we collectively refer them as regional floods by explicitly highlighting their spatial extent, that is over a majority of basins within a

neighborhood rather than over individual isolated basins (i.e., termed as local floods). Understanding the processes and controls of regional floods is motivated by the mounting evidence of increased spatial extents of extreme rainfall under a warming climate (Chen et al., 2023; Dai and Nie, 2022; Tan et al., 2021) and the resultant large-scale flood hazards over several continental regions, e.g., Europe (Kemter et al., 2020; Berghuijs et al., 2019), East Asia (Yang et al., 2022), and South Asia (Roxy et al., 2017).



The nature of regional floods that vary both spatially and temporally makes conventional site-specific flood frequency analyses less robust in estimating their hazardous potentials (Timonina et al., 2015; Neal et al., 2013; Brunner et al., 2019). The estimation bias can be especially prominent for floods with large return periods (Nguyen et al., 2020; Metin et al., 2020). Therefore, accurate flood risk assessment requires characterizing the spatial dependence of floods (i.e., the extent to which floods co-occur at different nearby locations) rather than identifying them as isolated local floods. Existing endeavors

principally rely on multivariate statistical models (Keef et al., 2009b; Heffernan and Tawn, 2004; Keef et al., 2013; Brunner et al., 2019; Lamb et al., 2010), numerical model chains (Falter et al., 2015) or combining both physical and statistical models (Quinn et al., 2019; Neal et al., 2013). For instance, Brunner et al. (2019) conduct multivariate frequency analyses using the copula theory, and show contrasting flood risk estimates from those based on conventional site-specific approaches.

      There are data-driven approaches in characterizing regional floods and their resultant impacts. For instance, Uhlemann

et al. (2010) identify regional floods through selecting flood peaks larger than local 10-year flood within a time window. They characterize flood severity by proposing a metric depending on stream orders. Similarly, Lu et al. (2017) considers the role of drainage networks in regional flood processes. They evaluate regional flood severity relying on empirical distributions of flood ratio (i.e., ratio of flood peak discharge to the sample 10-year flood discharge). Brunner et al. (2020b) define regional floods as the co-occurrences of site-specific flood peaks with similar ranks and sufficiently large magnitudes. They further

characterize the degree of spatial dependence of floods according to the number of concurrent flood peaks. Tarouilly et al. (2021) identify regional floods by picking up basins with flood peak discharge exceeding certain threlhods (similarly also see Brunner et al., 2022). Exisiting approaches, however, do not explicitly require regional floods to be spatially contiguous but only focus on whether their occurrences are within a small time window or not. This may not be a problem if the setting of interest is a moderate-sized basin or a small region with limited hydrological heterogeneities (e.g. Brunner et al., 2020a).

Berghuijs et al. (2019) try to remedy this issue by characterizing regional floods with concurrent flood peaks over a prescribed shape (i.e., circle in their case) of bufferring regions within which there are at least 50 % stations experiencing floods. Based on the notion of image connectivity, Wang et al. (2023) identify contiguous quantities of runoff grids in both space and time as regional floods. Due to the regular grid spacings of simulated runoff fields, there is no need to prescribe either the shape of the flood extent or the ratio of grids experiencing floods. This advantage unfortunately, cannot be inherited by in-situ stream

gauging observations.

      The spatial dependence of floods are related to large-scale weather systems (Villarini et al., 2011), land-surface processes (Brunner et al., 2020b; Lu et al., 2023) and hydraulic structures (Turner-Gillespie et al., 2003; Brunner, 2021). Brunner et al. (2020b) show that the spatial dependence of floods varies with season and region, with winter and spring showing the largest spatial dependence and thus the highest widespread flooding potential over US. They show that the spatial dependence of

rainfall does not always translate into floods due to the disturbance of land-surface processes (i.e., soil moisture dynamics, snowmelt). Tarouilly et al. (2021) show that regional floods over western US are mainly induced by extreme rainfall associated with atmospheric rivers in winter, snowmelt in spring and tropical storms in summer, but the most extreme floods reflect the





combination of both intense rainfall and favourable land surface processes (e.g., snowmelt). Nanditha and Mishra (2022) confirm their results by further showing that heavy rainfall on wet soils is a prominent driver for large-scale flooding over the Indian river basins. Elevated soil moisture can be induced by snowmelt or excessive rainfall. This is believed to have contributed to flood intensity more than different storm types (Brunner and Dougherty, 2022). Keef et al. (2009a) find negligible impacts of lakes and reservoirs on the spatial dependence of floods in Great Britain. Brunner (2021) show that spatial dependence of floods is reduced by reservoirs in winter and fall across US, but varies in spring and summer depending on catchment regulation measures.

Relative importance of meteorological forcing and land surface processes for regional floods over the monsoon regions, such as eastern China, has not been elucidated. This is challenged by the mixture of precipitating systems (e.g., monsoon fronts, tropical cyclones, extratropical cyclones, etc.) and the resultant rainfall variabilities in both space and time. Existing evidence over eastern China show contrasting behaviors of regional floods, in terms of spatial extent and intensity. For instance, the July 1931 flood over Yangtze River, with approximately 180,000 $km^2$ inundated areas and 2 million fatalities (Buck, 1932), is a "poster child" of the deadliest widespread flood hazards in the world. Extreme rainfall and flood inundation submerged eight provinces over eastern China (Zhou et al., 2023). Another example is the August 1975 flood over the Huai River that resulted in less than 1/3 inundation area as the July 1931 flood but comparable economic losses (Qing et al., 2016). The August 1975 flood is also responsible for the world's 6-hour rainfall record (i.e., 830 mm) and several unit peak discharges on the world's flood envelop curve (Yang et al., 2017). Understanding processes and controls of regional floods over eastern China, especially pertaining to their contrasting behaviors, can serve as the basis for large-scale flood hazard modeling and risk assessment.

Yang et al. (2021a) shows that extreme floods over East Asian summer monsoon region tend to cluster in the topographic-transition regions along Mt. Qinling and Mt. Taihang (i.e., the north portion of the topographic divide over eastern China, see the map in Appendix). Since some of those flood samples "define" the world's flood envelop curve, it remains unsettled about the spatial extents of these extremes. We hypothesize that extreme floods occur simultaneously with neighboring basins as regional floods, rather than as a local flood (that is in isolation with their neighbors).

Based on the aforementioned knowledge gaps, we propose an innovative framework for regional floods analyses that relies on in-situ stream gauging observations over eastern China. The core of the framework is to identify regional floods using the Density Based Spatial Clustering Applications with Noise (DBSCAN) algorithm. We develop a series of metrics to quantitatively characterize the spatial extent, magnitude and potential impacts of regional floods. We further shed light on the controls of the contrasting flood behaviors (in terms of spatial extents and magnitudes) by establishing a statistical model between flood metrics and potential explanatory variables. We expect to advance the characterization of flood hazards by highlighting their spatial extents.

Our empirical analyses are centered on the following research questions: (1) what are the spatial and temporal patterns of regional floods over eastern China? (2) do extreme floods cluster in space and time? (3) what are the key ingredients of





flood-producing storms for large-scale flood hazards? (4) how do rainfall forcing and land surface processes determine the contrasting behaviors of regional floods over eastern China?

## 2 Data and Methods

### 2.1 Dataset

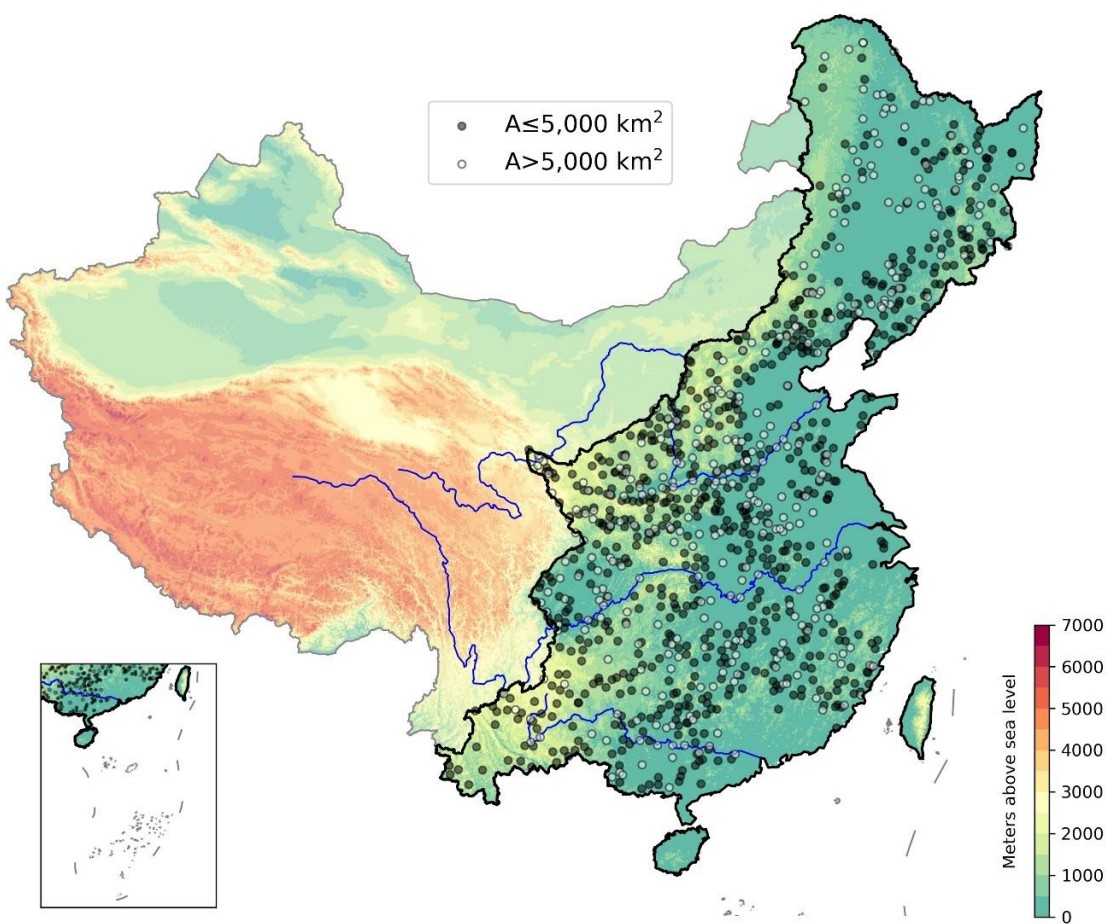

Figure 1. Overview of stream gauging stations (all dots, 1,036 in total) over the East Asian monsoon region (outlined by the dark black line, Liu & Shi, 2015). Grey and white dots show stations with drainage areas smaller and greater than 5,000 km$^2$, respectively. Three blue lines from north to south represent the Yellow River, Yangtze River and Pearl River, respectively. Shading represents elevation in meters above sea level. The inset plot represents the southernmost Chinese territory.





105       Our analyses are based on a dense network of stream gauging stations over eastern China (Fig. 1). There are 1,036 stream gauging stations in total. Each of them has at least 35-year observational records of annual maximum flood peaks (AMFs, including magnitude and timing) during 1980-2017 (Fig. 1). The number of complete-record stations remains constant throughout the period. The continuous flood observations have gone through strict quality controls by following standard procedures (including the removal of enormously large values or constantly low values during the period, etc.). This dataset

has been used in previous flood studies (e.g., Yang et al., 2019; Yang et al., 2021b). The drainage areas of these gauges range from 10 to 1.7 million $km^2$, with 70 % of them less than 5,000 $km^2$. The drainage boundaries are delineated using Hydrological data and maps based on SHuttle Elevation Derivatives at multiple Scales (HydroSHEDS) in ArcGIS, and are then checked against the archive maintained by the Ministry of Water Resources, China.

      Our empirical analyses of rainfall and soil moisture are based on the gridded CN05.1 daily rainfall product and the Fifth

Generation of European Reanalysis (ERA5) hourly soil moisture (i.e., the top soil layer at 0-7 cm) dataset over mainland China. The gridded CN05.1 rainfall product is interpolated from in-situ observations of 2,416 rain gauges across China (Wu and Gao, 2013). The hourly soil moisture dataset is resampled to daily scale by totaling hourly records within a calendar day (that is from 00 UTC to 00 UTC of the following day). The spatial resolution of both products is 0.25°. Both rainfall and soil moisture datasets have been validated over mainland China with good performance (Wu and Gao, 2013; Li et al., 2020b; Li et al., 2020a;

Sun et al., 2021).

### 2.2 Analytical framework of regional floods

      Our proposed analytical framework of regional floods includes four parts: (a) identification, (b) characterization, (c) categorization, and (d) statistical modeling. We demonstrate the workflow in Fig. 2. We provide a flowchart in Fig. S1 to shows sequential steps of data input, data pre-process and processing.

*(a) Identification*

      We define a regional flood (referred to as *RegFl*) based on its intrinsic definition, that is a *RegFl* represents multiple flood peaks over several neighboring basins that occur within a certain period of time. We term flood peaks that are isolated from other flood peaks either in space or time as isolated floods (referred to as *IsoFls*). The procedures to identify a *RegFl* are as follows.

130       We first set a moving time window of *T* days, and pick up stream gauging stations with observed AMFs during the *T*-day time window (Fig. 2a). Here we set *T* to 15 days. The choice of a 15-day time window is designed to capture the entire rainfall and flood-generation processes for a wide range of drainage basin sizes (Boyd, 1978). We then apply a machine-learning based algorithm, DBSCAN, to automatically cluster those picked stations into a set of spatially contiguous clusters according to their geographic locations (Fig. 2b and 2c). We choose DBSCAN because it is designed to identify clusters with

arbitrary shapes as well as outliers based on samples' density (Ester et al., 1996). The algorithm does not require a pre-defined number of clusters compared to other clustering methods (such as K-means). There are two hyperparameters in the algorithm,





i.e., neighborhood scale $\varepsilon$ and the minimum number of points *MinPts*. We determine the two hyperparameters through the K-Nearest Neighbor (KNN) approach (Ester et al., 1996), where $\varepsilon$ is determined through detecting the "knee" of the KNN plot, while *K* equals to the value of *MinPts*. The "knee" represents the closest point to the origin of the plot (Fig. 2b). *MinPts* can

be interpreted as how many samples are required to define a neighborhood within which the sample density can be evaluated. Smaller *MinPts* identify clusters with less dense cores. In this study, *MinPts* is set to 10. This is determined by manually checking the clustering results (i.e., flood extent) with different *MinPts* against the corresponding spatial patterns of heavy rainfall and historical flood records (e.g., maps, documents). Further systematic validation is carried out by comparing against an independent flood archive (see details below). To obtain a reliable KNN plot and the "knee", we require at least *M* samples

(i.e., picked stations). We set *M* to 50. Larger *MinPts* (i.e., *MinPts* =15 or 20) or different *M* values (i.e., *M*=40 or 60), show little impacts on our results. The choice of the two hyperparameters depends on how samples are spatially distributed as well as the overall densities. The spatial pattern of identified clustered remain unchanged by selecting a subset of gauges with relatively uniform distributions, indicating negligible impact of gauge density (Fig. S3).

The 15-day time window moves from the first to the last date of AMF occurrences for each year. We thus obtain all

qualified clusters (i.e., termed as potential *RegFls*). We use the smallest convex-hull polygon that bounds all AMFs to represent the flood footprint. Due to the propagation of precipitating systems, the extent and position of footprint changes with time (Fig. 2d). We keep the largest convex-hull polygon representing the largest flood footprint during the time window. Those smaller polygons, representing the developing or decaying flood pulses, are removed. The final selection of the largest polygons constitutes our *RegFls* catalog. A smaller time window (e.g., *T*=7) identifies more *RegFls* but a consistent spatial-temporal

pattern with that of using a 15-day time window. This is because flood-producing storms are separated into multiple episodes, rather than treating them as consecutive.

We verify the capability of our algorithm in representing large-scale flood hazards by comparing our *RegFls* catalog against the Dartmouth Flood Observation dataset (DFO, Brakenridge, 2016). DFO provides details of observed flood hazards (including their dates of occurrences, spatial extents, and socioeconomic impacts) from 1985 till the present year from

miscellaneous sources (e.g., newspaper, observations, satellite images, etc.). It has been widely used as a benchmark for other flood datasets (Wang et al., 2023; Tellman et al., 2021; Dottori et al., 2016) and flood hazard modeling analyses (Kron et al., 2012; Carozza and Boudreault, 2021). We choose DFO over other state-of-the-art flood datasets due to its record length that largely overlaps with our dataset (e.g., compared to the Global Flood Database, Tellman et al., 2021) and its details in documenting flood spatial extents (compared to the Emergency Events Database, Guha-Sapir et al., 2016).





Figure 2. Schematic plot for the analytical framework of regional floods.






*(b) Characterization*

We characterize the spatial extent, intensity, and severity of a *RegFl* based on a series of gauge-based metrics. Only
basins with drainage areas less than 5,000 km$^2$ are considered in the characterization, to avoid the impact of nested basins as much as possible.

We characterize the spatial extent of a *RegFl* based on the cascade-union area of all watersheds that constitute the flood (Fig. 2e). It basically represents the largest drainage area for all non-nested watersheds within the *RegFl*. An alternative way to represent the spatial extent is based on a convex-hull polygon (see Fig. 2d for example). The spearman correlation coefficient
between the coverages of the cascade-union watershed and the convex-hull polygon is 0.87 (*P*<0.001). They are significantly correlated with the number of AMFs (*P*<0.001).

The severity of a *RegFl* represents the accumulative impacts of multiple floods, while the intensity represents average flood peak magnitude. To make the characteristics of different *RegFls* comparable, we need to normalize AMF magnitudes. This is because AMF vary drastically across drainage basins. Here we use inversed rank of each AMF across its observational
period (Fig. 2f, similarly see Tarouilly, Li et al., 2021). There are other ways of normalizing flood peaks, based on, e.g., unit peak discharge (i.e., ratio of flood peak magnitude to drainage area, e.g., Herschy, 2002; Li et al., 2013) or flood ratio (e.g., Smith et al., 2018). Those metrics tend to be biased towards either small drainage basins (for unit peak discharge) or basins with heavy tails of flood peak distributions (for flood ratio, see Yang, Yang et al., 2021a) . We note that inversed rank does not show dependence on either drainage area or tail property of flood peak distributions (Fig. S2).

The severity of a *RegFl* is contributed by both the magnitude of individual AMFs (i.e., inversed rank) and their spatial extent (i.e., consistent with the number of AMFs within the cluster). The severity of a *RegFl* can then be simplified as the summation of the inversed ranks for AMFs in the *RegFl*.

$$RFI = \sum_{i=1}^{i=N} \frac{1}{Rank_i} \tag{1}$$

where *RFI* represents *RegFls* severity, $Rank_i$ represents the rank of *i*th AMF within its observational records. *N* represents the
total number of AMFs clustered in a *RegFl*. The averaged *RFI* over all AMFs, i.e., mean severity, is used to represent the intensity of a *RegFl*. A possible caveat of *RFI* is that it cannot effectively distinguish extreme floods from moderate ones. Since our intention is to characterize flood hazards at regional scale, the accumulative inversed rank is able to differentiate the potential of regional flood hazards over multiple neighboring basins. We use flood ratio occasionally to highlight the most extreme floods in the following analyses.

*(c) Categorization*

Intuitively, there are floods with large spatial extent but relatively less intensity (e.g., the 1931 Yangtze River flood), while floods with the opposite combinations also exist (e.g., the 1975 Huai River flood). We categorize *RegFls* into different groups according to their spatial extent and intensity, to highlight distinct processes that determine hazardous potentials.





We adopt the K-means algorithm for the categorization. We choose K-means due to its simplicity and easy
interpretability (Everitt et al., 2011). We standardize both spatial extent and intensity of each *RegFl* before clustering, i.e.,
extracted by the mean and divided by the standard deviation. The optimal number of clusters is automatically determined based
on the Silhouette score (Rousseeuw, 1987) and Davies-Bouldin score (Davies and Bouldin, 1979). The clustering results agree
with intuitive understandings of flood hazards over eastern China, justifying the choice of K-means over other algorithms (Fig.
2g).

*(d) Statistical modeling*

Finally, we establish statistical models between characteristics of *RegFls* (i.e., severity and extent) and their potential
explainable predictors, to shed light on the controls of regional flood processes (Fig. 2j). We first extract basin-average annual
maximum rainfall and antecedent soil moisture at various temporal scales (i.e., 1-day, 3-day, 5-day, and 7-day) for each basin
within the identified cascade-union region. The basin-average rainfall and soil moisture is first normalized by dividing its local
annual 75th percentile, and then summed to represent different atmospheric conditions as well as antecedent wetness. We use
the fractions of different land use/land cover over the period 1980-2015 for each five years (i.e., the value from the closest
year to the date of flood occurrences), the mean slope and the total number of dams within the cascade-union watersheds to
represent physiographic attributes (Fig. 2e and Table 1). These selected predictors have been previously verified with notable
impacts on basin-scale flood responses (e.g. Liu et al., 2015; Hall et al., 2014). These predictors are by all means not exhaustive,
but potentially represent potential drivers responsible for flood-generation processes.

We adopt the Conditional Random Forest (CRF) model as the statistical modelling tool. The CRF model is a variant of
the random forest model (Zeileis et al., 2008; Strobl et al., 2008). The model is able to provide unbiased conditional variable
important measures for correlated predictor variables (Tyralis et al., 2019). Each explainable predictor is referred to as a feature
in the model. There are two key hyperparameters, *ntree* and *mtry*, where *ntree* decides the number of times bootstrap samples
are generated, and *mtry* represents the number of predictor features selected as candidates for tree splitting. In this study, *mtry*
ranges from 2 to 11, while *ntree* varies from 50 to 500 with an interval of 50. We evaluate the model performance using out-
of-bag (i.e., samples left after bootstrapping) rooted mean square error (RMSE) and coefficient of determination (i.e., R-
squared). The best combination of *ntree* and *mtry* is determined when RMSE is the smallest (see Table S1 for evaluation
metrics). Non-parametric statistical tests are used to test differences between training error and out-of-bag error for each model,
to shed light on whether there is any overfitting (see Table S1 for details). We use the conditional permutation feature
importance to evaluate the importance of each explainable predictor (Debeer and Strobl, 2020). This metric takes care of
mutual correlation of predictors by introducing a conditional permutation scheme (Strobl et al., 2008).






Table 1. Summary of potential explainable variables in predicting *RegFls* characteristics.

| Type | Variable | Full name | Data sources |
|---|---|---|---|
| Rainfall | P1dmax | maximum 1-day rainfall | The gridded CN05.1 daily rainfall product, with a spatial resolution of 0.25° |
| | P3dmax | maximum 3-day rainfall | |
| | P5dmax | maximum 5-day rainfall | |
| | P7dmax | maximum 7-day rainfall | |
| Antecedent soil moisture | SM_1d | antecedent 1-day soil moisture | The hourly ERA5 soil moisture product, with a spatial resolution of 0.25°. The hourly scale is resampled into daily scale by summation. |
| | SM_3d | antecedent 3-day soil moisture | |
| | SM_5d | antecedent 5-day soil moisture | |
| | SM_7d | antecedent 7-day soil moisture | |
| Physiographic | SlopePCT | average slope | SRTM dataset, with a spatial resolution of 90 m. |
| | LakePCT | fraction of lake coverage | Global HydroLAKES dataset |
| | DamCount | number of dams | Ministry of Water Resources, China, including reservoirs with capacity exceeding 10 million $m^3$ |
| | UrbanPCT | fraction of urban land coverage | The 1-km RESDC-CAS land use dataset for the year 1980, 1985, 1990, 1995, 2000, and 2005. |
| | ForestPCT | fraction of forest land coverage | |
| | CropPCT | fraction of crop land coverage | |

## 2.3 Empirical analyses

We highlight regional flood processes based on empirical analyses of rainfall and soil moisture anomalies within the 15-day time window. We extract and composite time series of daily basin-average rainfall and soil moisture 7-days before and after each AMF within a *RegFl* (Fig. 2h). We do the composition by placing the date of flood peak occurrence in the center for both the rainfall and soil moisture series. We normalize the series by dividing the annual 75th percentile daily rainfall (over rainy days, with daily rain rate exceeding 0.1 mm/d) and the annual 75th percentile daily soil moisture (over days with soil

moisture greater than 0 $m^3/m^3$), respectively.

We examine the fine-scale structures of flood-producing storms. We first label all the CN05.1 rainfall grids with rain rate exceeding the annual 75th percentile daily rainfall. We then identify all spatially continuous patches from grids. Each patch is then given an identifier, and termed as an individual storm cell. All the storm cells that overlap with each drainage basin within the 7 days prior to the day of each AMF are deemed as flood-producing storms (Fig. 2i). This is implemented for

*RegFls* of different groups (see Section 2.2c). We compare statistics of storms cells (including the total number, mean size, and mean orientation) across different *RegFls* groups, to highlight the fine-scale structures of flood-producing storms.





Landfalling tropical cyclone (TC) is an important flood-producing agent over eastern China (Yang et al., 2020). We associate a TC with *RegFls* by making a 300-km buffer centered around TC track (Gaona et al., 2018). If there is any intersection between the buffer zone and the convex-hull polygon of a *RegFl* during the 15-day time window, we label the

*RegFl* as a TC-induced *RegFl*. We use convex-hull polygon to associate each *RegFl* with TCs, since it is a spatially contiguous quantity with which to associate large-scale meteorological drivers. TC tracks are provided by the International Best Track Archive for Climate Stewardship (IBTrACS), with records of longitude and latitude of TC center as well as its nature (e.g., tropical storm, tropical cyclone, extratropical transition) at a 6-hour time interval.

## 3 Results and Discussions

### 3.1 Overview of the regional flood catalog

We identify 318 *RegFls* during the period of 1980-2017 over eastern China, i.e., approximately 8.3 per year on average. These *RegFls* consist of 22,902 AMFs, accounting for around 55 % of the total AMFs (i.e., the accumulated number of years for all stream gauging stations over eastern China). There are 72 AMFs on average for each *RegFl*, with the number ranging from 6 to 317. The remaining 45 % AMFs are not clustered into any *RegFls*, and are termed as *IsoFls*. These *IsoFls* are either

remote in space (beyond 1,000 km on average) or induced by isolated storms (beyond 15 days), and cannot be identified as occurring in any spatially contiguous regions.

We compare our *RegFls* catalog against the DFO dataset. There are 274 floods observed in both DFO and our catalog during the overlapping period, i.e., 1985-2017. More specifically, 53 % of the DFO floods can be well captured by our catalog, with the DFO flood extent enveloped by the convex-hull polygons (i.e., see Section 3.2 for event comparisons). The missing

representation of DFO floods by our catalog can be partially related to the limitation that only AMFs are adopted in our *RegFls* identification. The comparable spatial patterns of the DFO floods and our *RegFls* catalog demonstrate capabilities of our proposed framework in examining regional flood hazards.

### 3.2 Spatial-temporal patterns of regional floods

*RegFls* are spatially clustered over eastern China, with northeastern China, central China, and southern Yangtze River

as three hotspots (Fig. 3). There are more than 30 *RegFls* per river reach within the hotspots. These hotspots are distributed over complex terrains (see Fig. A1 for details), highlighting the role of topography in dictating severe large-scale flood hazards over China. The spatial clustering is closely tied to the properties of flood-producing systems (in terms of their propagation speed and intensity) as well as their interactions with regional topography. For instance, the monsoon front propagates over the middle Yangtze River basin around June. The front tends to remain stagnant for a while before abruptly jumping to central

and northern China in middle July. Excessive rainfall during the stagnancy is responsible for frequent *RegFls* in the southern Yangtze River (HS3), but less possibility of *RegFls* over the transitional region (i.e., in between HS2 and HS3, Fig. 3). The





elevated rainfall intensity (i.e., orographic lifting) and stagnant storm motion (i.e., topographic blocking) collectively lead to a temporal clustering of extreme rainfall in central China. This explains why central China experiences the most frequent *RegFls* and some of the severest flood hazards (see detailed analyses below).

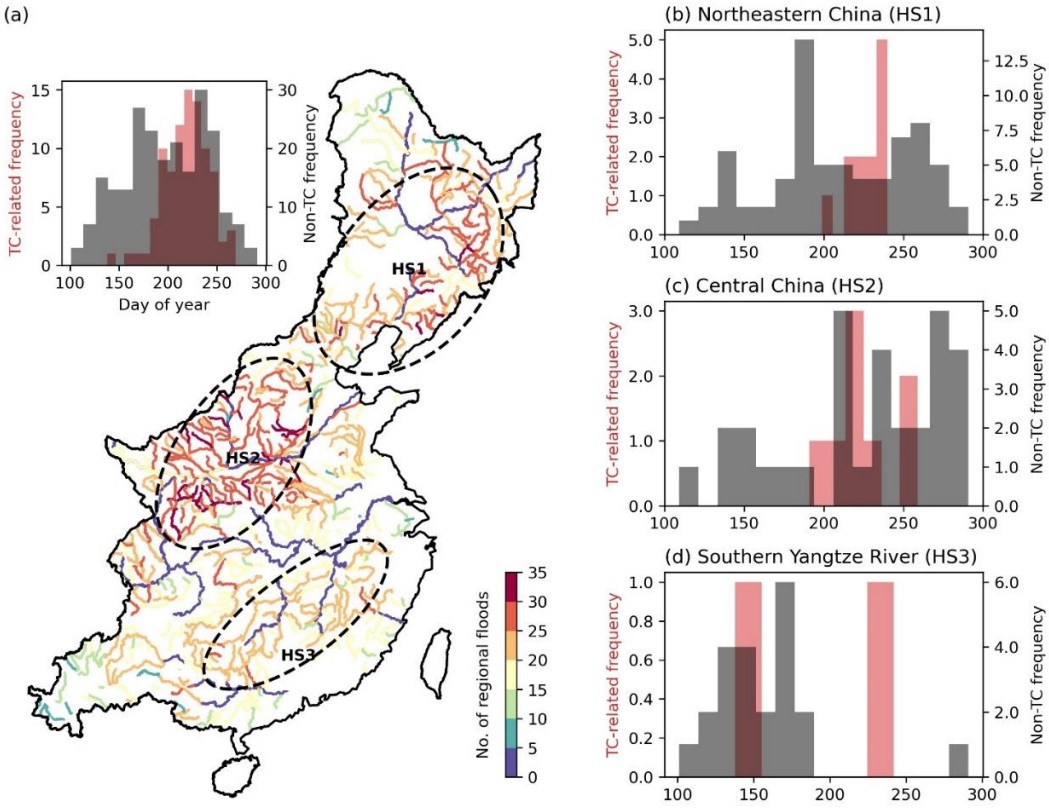

Figure 3. (a) Spatial-temporal patterns of *RegFls* during 1980-2017. The inset bar plot shows the temporal distribution of the middle date of TC-related (red) and Non-TC (blue) *RegFls*. The frequency of *RegFls* for each river reach is estimated by matching AMFs (drainage area smaller than 5,000 km$^2$) in *RegFls* with the river reach using a 500-meter buffer. Three black dashed-line circles highlight the three hotspots (HS). (b)-(d) Distribution of the middle date of *RegFls* in northeastern China (HS1), central China (HS2) and southern Yangtze River (HS3), respectively.

There are one third *RegFls* associated with landfalling TCs over eastern China. A notable feature is that TC-induced floods show striking temporal clustering, with 77 % occurred within a two-months period, i.e., from early July to late August (Fig. 3). This is contrary to *RegFls* induced by other precipitating systems (e.g., monsoon front, extratropical cyclones, etc.) that spread across warm season. The temporal clustering is mainly regulated by the behaviors of TCs genesis over the Western North Pacific basin. Interactions of landfalling TCs with regional topography (e.g., southern Mt. Taihang and Mt. Qinling) is a key ingredient for *RegFls* over eastern China (e.g., the August 1975 Huai River flood).



**(a) Frequency of Regional Floods (RegFls)**

**(b) Seasonality of Regional Floods (RegFls)**

**(c) Frequency of Isolated Floods (IsoFls)**

**(d) Seasonality of Isolated Floods (IsoFls)**

Figure 4. Frequency and seasonality of regional (a-b) and isolated (c-d) floods. Circular statistics are applied to obtain the mean date of occurrence of AMFs for each station (refer to (Berens, 2009; Pewsey et al., 2013; Blöschl et al., 2017) for details of circular statistics).

Unlike *RegFls*, *IsoFls* over eastern China are less clustered in space or time. We note that southern China is more likely to experience *IsoFls* than its northern counterpart, with slightly higher frequencies along the main stream of the Yangtze River

(Fig. 4a and 4c). *IsoFls* are more uniformly distributed across the warm season (i.e., April to October) compared to *RegFls* (Fig. 4 and Fig. 5). *RegFls*, however, tend to show greater temporal clustering (Fig. 5a and Fig. S4) and have much larger flood peak magnitudes than *IsoFls* (Fig. 5b). Approximately two thirds of the record floods (i.e., the largest flood for a station during its entire observational record) are observed in *RegFls*. Non-parametric Mann-Whitney U test and Kruskal-Wallis test suggest that the flood ratios between two groups are statistically different (both *P*<0.001), with *RegFls* being larger. This indicates that

extreme floods tend to be concurrent with neighboring basins rather than isolated in space and time. This is likely dictated by the space-time organizations of precipitation systems (e.g., spatial extents, duration) and/or their interactions with regional topography over East Asian summer monsoon region.

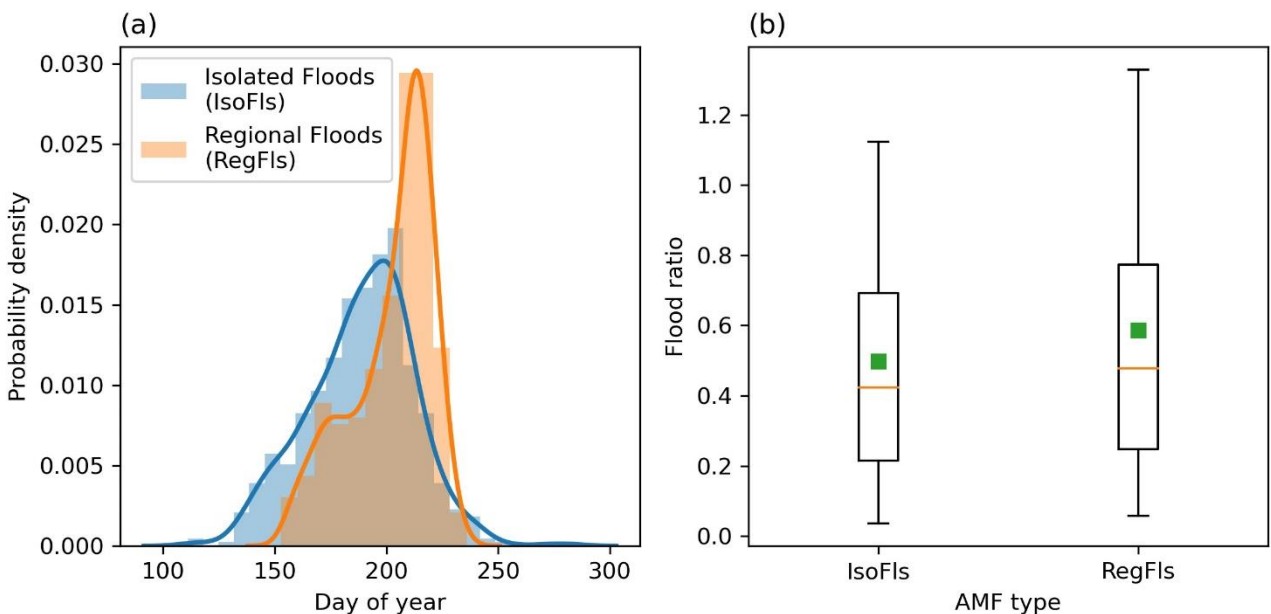

Figure 5. (a) Distribution of circular mean date of isolated (blue) and regional (orange) floods. (b) Boxplots of flood ratio for
isolated floods (*IsoFls*) and regional floods (*RegFls*). The orange line and green square within the box represent the median and mean values, respectively. The box spans the 25th and 75th percentile, and the whiskers represent the minimum and maximum values.

**3.3 Synoptic processes of regional floods**

315   Figure 6 shows the spatial maps of top twelve severest *RegFls* (based on the severity index *RFI*) over eastern China (see Appendix for synoptic descriptions). A notable feature is that central China (i.e., the middle-lower Yellow River region) is affected by all twelve *RegFls*. The occurrence frequency of *RegFls* in this region is also the largest, with 25 *RegFls* per stream gauging station on average during the 38-year period.



Figure 6. The top twelve most severe *RegFls* over eastern China. Blue polygons represent the spatial extents of the DFO floods. Black dashed-line polygons represent the convex-hull region of *RegFls*. Small red polygons inside the convex-hulls represent watersheds with AMFs (i.e., shade representing ranks). Grey points represent stations without AMFs. Grey solid lines represent TC tracks. The middle date of occurrences for each *RegFl* and the associated *RFI* is shown in the subtitles.

The severest *RegFl* occurs during July 2016, with the *RFI* equal to 39.4 (Fig. 6a). Torrential rainfall during 18 July 2016 to 1 August 2016 lead to 287 AMFs across central and northern China. The flood is directly responsible for 130 fatalities and substantial socio-economic losses (Lei et al., 2017). There are 36 AMFs (i.e., 13 % of all AMFs in this *RegFl*) with their magnitudes larger than the sample 10-year flood, i.e., the flood ratio larger than 1.0. The maximum flood ratio is 13.4. It is the



sixth largest flood ratio during 1980-2017 over eastern China. The *RegFl* document eight record floods. Rainfall intensity
exceeds 20 mm/h over a large portion of the flood region. The anomaly of rainfall accumulation is 300 mm larger than the
climatological mean (Fig. S5). Extreme rainfall for the July 2016 flood is tied to anomalous position of the Western Pacific
Subtropical High that extends westward onto the east Asian continent. The synoptic configuration facilitates moisture transport
from western Pacific onto eastern China along its southern fringe (Yuan et al., 2017). The year 2016 is also the wettest flood
season during the past six decades (Gao et al., 2018). Abnormally large rainfall intensity superimposed on notably wet soils
(i.e., twenty times as large as the climatological mean state, Fig. S6) collectively contributes to the severe flood hazards.

As expected, landfalling TCs are responsible for some of the severest *RegFls*. For instance, the three TCs, i.e., Hope
(1989), Herb (1996), and Toraji (2001), are responsible for the three out of the top five severest *RegFls*. These TCs underwent
extratropical transition processes (Fig. 6), and are comparable in their tracks, with similar patterns of flood footprint as well.
An interesting finding is that flood regions are mostly located beyond the termination of TCs tracks (see the black dashed line
in Fig. 6). This highlights the potential of TC remnants in producing severe flood hazards over eastern China (similarly see
Smith et al. (2023).

The northeast vortex is the most frequently recurring weather system responsible for *RegFls* (Table A1). It produces
persistent and widespread rainfall during the post-Meiyu period (Xie et al., 2015), and is responsible for 53 % extreme rainfall
in northeastern China (Tang et al., 2021). Except for Typhoon Kalmaegi (2014), almost all flood-producing TCs are
accompanied by northeast vortex (Table A1). There are six of the top twelve *RegFls* associated with southwest vortexes. The
cut-off lows, developed from an eastward-propagating westerly trough, are further responsible for five out of the top twelve
most severe *RegFls*. Synoptic analyses of flood-producing storms highlight the importance of cyclonic precipitating systems
(e.g., tropical cyclones, southwest vortexes, cut-off lows) in dictating large-scale flood hazards over the East Asian monsoon
region.

**3.4 Categorization of regional floods**

The severity index *RFI* is contributed by both the total number of AMFs (i.e., spatial extent) and the mean inversed rank
of AMFs in observational records (i.e., intensity). The spearman correlation coefficient between *RFI* and the corresponding
spatial extent and mean intensity is 0.91 (*P*<0.001) and 0.36 (*P*<0.001), respectively. The spearman correlation coefficient
between spatial extent and intensity is only 0.08 (*P*=0.15). This means that there are different types of *RegFls*, depending on
the relative dominance of spatial extent and/or intensity on *RFI*.

We categorize the 318 *RegFls* into different groups by considering spatial extent and intensity (see Section 2.2c for
details). The optimal number of clusters is 3 (see Section 2.3). We name the three *RegFls* groups as Mild *RegFls* (*N*=176),
Large *RegFls* (*N*=103), and Intense *RegFls* (*N*=39), according to their positions on the "intensity-spatial extent" space domain
(Fig. 7). Figure 8 shows the spatial distributions of different *RegFls* groups. The Mild *RegFls* and Large *RegFls* tend to occur
more frequently in northeastern and central China, while the Intense *RegFls* (i.e., large in intensity but small in extent) show





weak geographic contrasts. The Large *RegFls* temporally cluster in early August, while the other two *RegFls* groups show a bimodal seasonal distribution. The temporal clustering of Large *RegFls* might be associated with frequent TCs genesis in the northwestern Pacific basin. There are 40% Large *RegFls* directly associated with landfalling TCs.

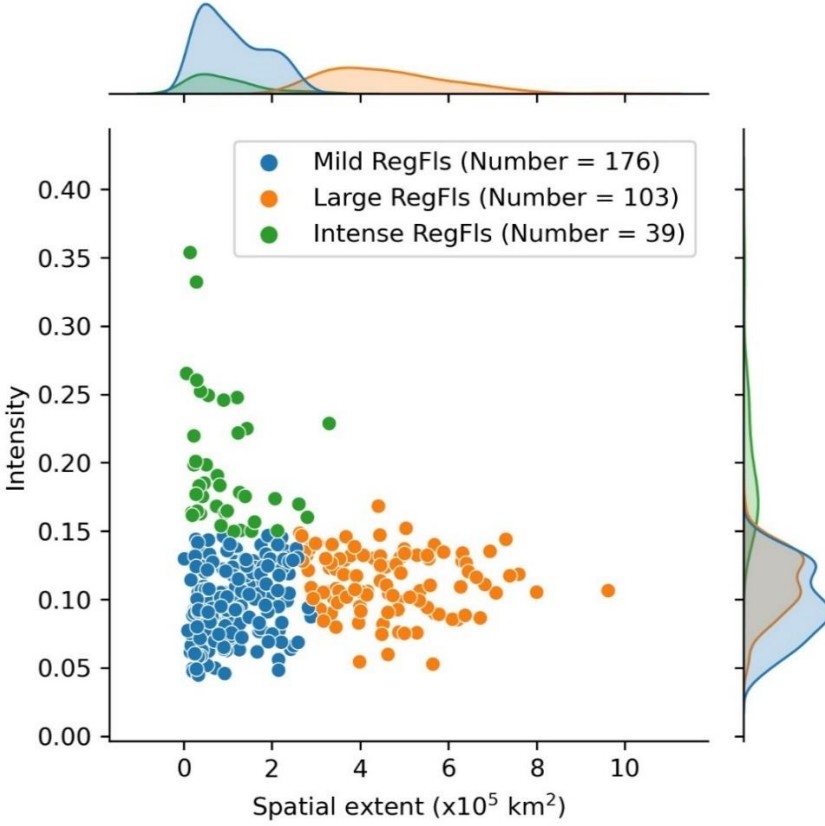

Figure 7. K-means classification of *RegFls* according to z-score of flood intensity (i.e., averaged inversed rank) and spatial extent (i.e., cascade-union watersheds areas). Blue, orange and green dots represent Mild, Large and Intense *RegFls*, respectively. Two subplots on the top and right corner show the probability density distribution of intensity and spatial extent.



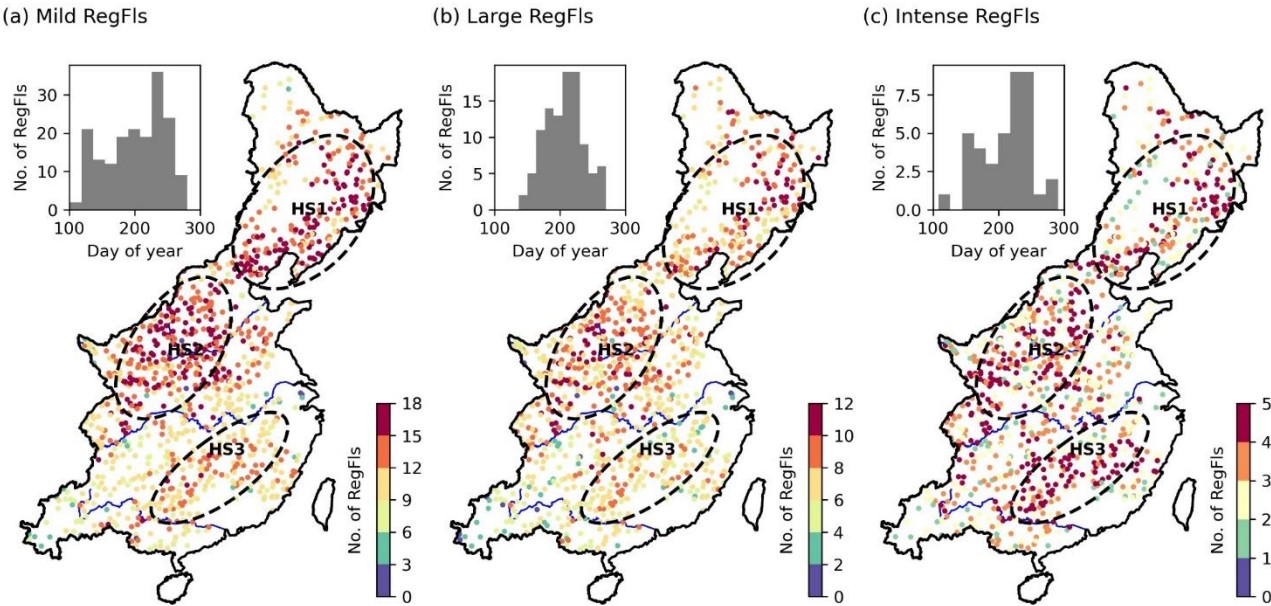

Figure 8. Spatial distribution of (a) Mild, (b) Large and (c) Intense *RegFls*. The inset bar plots show the temporal distribution of the middle date of flood occurrences. Three black dashed-line circles highlight three hotspots.

Contrasting flood behaviors from different *RegFls* groups are resulted from diverse regional-scale rainfall-runoff processes. Figure 9 shows the composite time series of daily rainfall and soil moisture for different *RegFls* groups (see Section

2.3 for details). There is a notable rainfall peak approximately one day earlier than flood peaks on average. Changes in daily rainfall show abrupt rising and falling limbs, with the composite mean rainfall peak approximately 1.5 times as large as the 75th percentile daily rainfall (Fig. 9a). However, there are negligible differences among the three *RegFls* groups in the composited rainfall series. The composited mean soil moisture is consistently above the local 75th percentile daily soil moisture. Unlike rainfall, the composited soil moisture shows notable differences among the three groups. For instance, the composited

mean daily soil moisture for Large *RegFls* is consistently larger than the other two groups. Similar contrast is also evident by only focusing on the 75th percentile soil moisture. Intense *RegFls* show slightly larger soil moisture content during its peak than Mild *RegFls* (Fig. 9b). This highlights the importance of antecedent soil wetness in dictating contrasting behaviors of *RegFls* over eastern China.

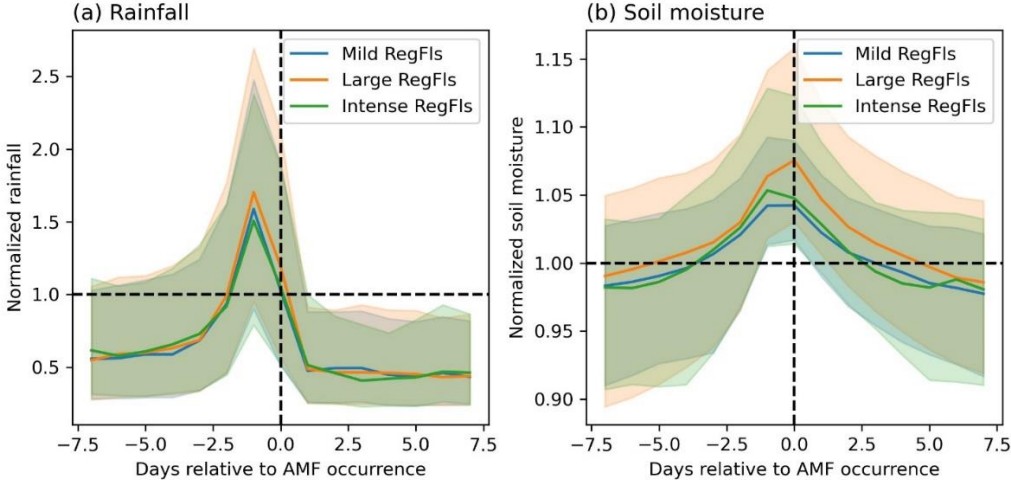

Figure 9. Lead-lag analyses of composite series of (a) rainfall and (b) soil moisture for three *RegFls* groups. The time series are extracted for each drainage basin within the *RegFl*, and are composited by placing the dates of AMF occurrences in the center. Shading represents range of 25th to 75th percentile.

Despite that the composited rainfall series are comparable, rainfall structures at fine-scale (see Section 2.3 for details)
show contrasting characteristics across different *RegFls* groups (Fig. 10). Large *RegFls* show the largest number of storm cells ($N$=6,238) but smallest storm size (i.e., with the median value of 72,500 km$^2$) (Fig. 11a and 11d). Intense *RegFls* show larger storm-averaged rainfall intensity (i.e., with the median value of 9.80 mm/h) than that of Large *RegFls* (i.e., with the median value of 7.38 mm/h) and Mild *RegFls* (i.e., with the median value of 9.37 mm/h, Fig. 11b). This indicates that Large *RegFls* are associated with a large number of small storm cells and relatively smaller intensities, while fewer but more intense storm
cells contribute to Intense *RegFls*. The median size of storm cells for Mild *RegFls* (225,312 km$^2$) and Intense *RegFls* (284,063 km$^2$) is comparable. These fine-scale storm features highlight the role of fine-scale rainfall organizations in distinguishing large-scale flood hazards.

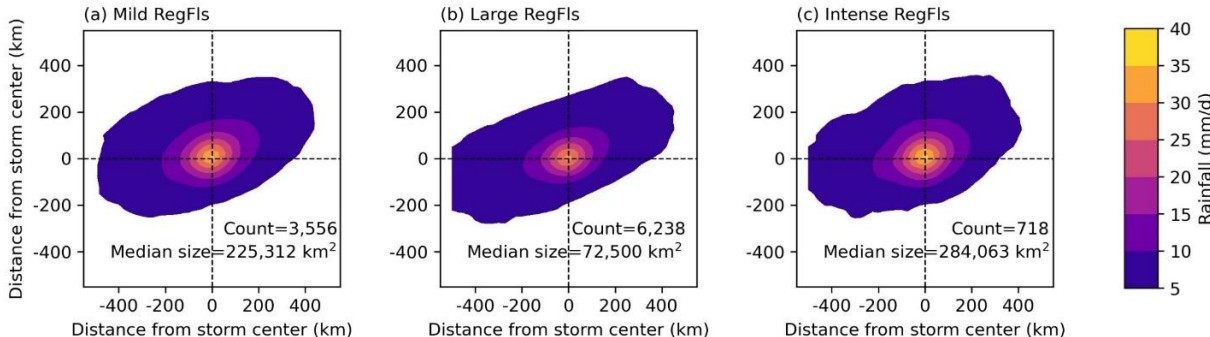

Figure 10. Composite storm cells for (a) Mild, (b) Large and (c) Intense *RegFls*. Shade represents daily rain rate (in mm/d).
The number of storm cells in the composite and their median size are also shown.

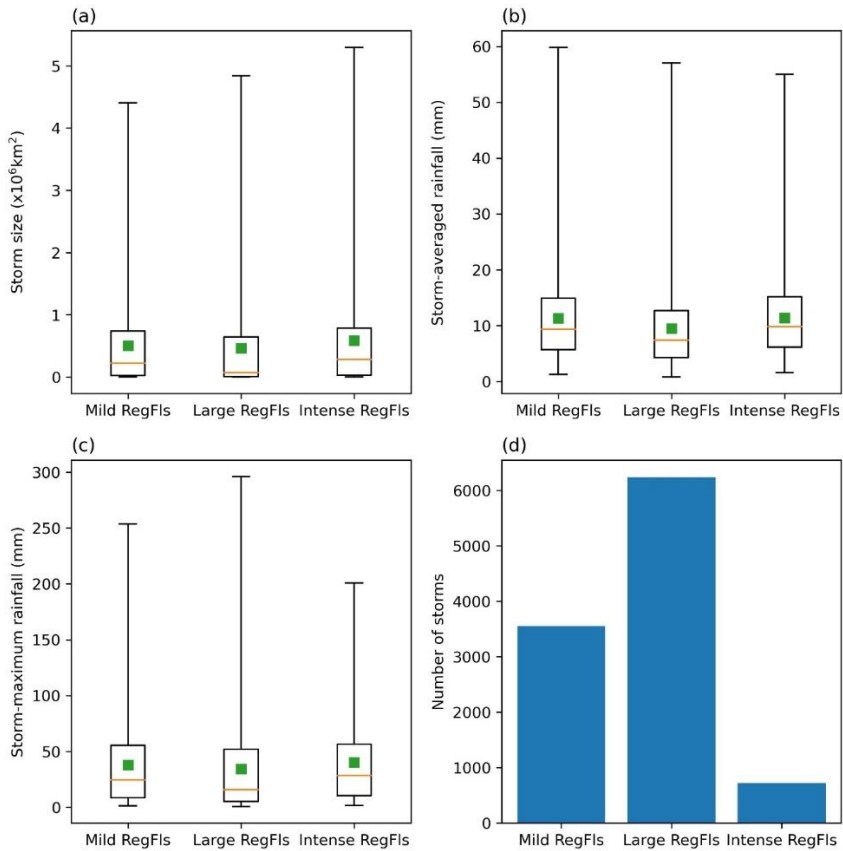

Figure 11. Statistics of flood-producing storms for the three *RegFls* groups. Boxplots of (a) storm size, (b) storm-averaged rainfall and (c) storm-maximum rainfall. (d) Total number of storm cells for the three *RegFls* groups. The orange line and

green square within the box represent the median and mean values, respectively. The box spans the 25th and 75th percentile, and the whiskers represent the minimum and maximum values.

**3.5 Statistical modeling of regional floods**

To further quantify the controls of contrasting flood behaviors, we establish CRF models between *RFI* (and spatial extent)

and potential explainable variables. A complete list of variables (i.e., features) is shown in Table 1. The out-of-bag RMSE for *RFI* is 2.36, ranging from 1.01 to 3.64 for three *RegFls* groups, while the coefficient of determination (i.e., R-squared) is 0.89, ranging from 0.43 to 0.86. This indicates that the selected explanatory variables can adequately explain the contrasting flood behaviors over eastern China. Similarly, we observe good model performance in determining spatial extents of *RegFls*, with





the out-of-bag RMSE and R-squared equal to 53,600 km$^2$ and 0.93, respectively. There is no significant difference between

training error and out-of-bag error, indicating weak evidence of overfitting (Table S1).

Antecedent soil moisture and maximum rainfall at 1-day and 3-day scales are the most important variables in determining *RFI* (Fig. 12). Physiographic attributes are less important, with dam counts and urban coverages posing only slight impacts. The variable importance is diverse across *RegFls* groups. More specifically, antecedent soil moisture (i.e., 3-day prior to flood peak) is prominent for Large *RegFls*, but basin-average rainfall at 1-day and 3-day stand out for Intense *RegFls*. This means

that large-scale floods are more likely to be triggered under wet soil conditions rather than contributed by local intense rainfall.

Figure 12. Conditional permutation importance of explanatory variables in predicting *RFI* of (a) All, (b) Mild, (c) Large and (d) Intense *RegFls*. Refer to Table 1 for details of the variables.





Antecedent soil moisture also stands out in determining contrasting the spatial extents of floods (Fig. S7). Neither physiographic attributes (e.g., percentage of different land use types, dam count) nor basin-average rainfall at various temporal scales show as comparable importance, except for Intense *RegFls* (Fig. S7). This is consistent with Section 3.3 that highlight contrasting soil moisture anomalies across different flood groups. Our results highlight the importance of soil moisture in dictating large-scale flood hazards. This is due to increased spatial dependence of floods under wet soil conditions, which is

further replenished by rainfall during the monsoon season.

## 4 Summary and Conclusions

In this study, we propose a machine-learning framework to investigate the processes and controls of *RegFls* over eastern China. Our analyses highlight distinct rainfall-runoff processes and drivers that dictate contrasting behaviors of *RegFls*. The main findings are summarized as follows.

(1) Identification of *RegFls*: Based on the new framework and a dense stream gauging network, we identify 318 *RegFls* over eastern China during 1980-2017. Our *RegFls* catalog provides detailed spatial-temporal characterization of large-scale flood hazards, and can serve as significant complement to existing flood datasets in the world.

(2) Spatial and temporal clustering of *RegFls*: *RegFls* are spatially and temporally clustered, with northeastern China, central China and southern Yangtze River as three hotspots with more frequent occurrences. The spatial clustering is dictated

by the propagation of precipitating weather systems (e.g., monsoon fronts, landfalling TCs) as well as their interactions with regional topography. The temporal clustering is associated with frequent landfalling TCs during late July to early August. TC remnants or extratropical transitions are important features of *RegFls* over eastern China. Cyclonic precipitating systems are frequent flood agents over the East Asian monsoon region.

(3) Isolated floods: *IsoFls* do not show either spatial or temporal clustering, compared to *RegFls*. The flood ratios of

*IsoFls* are statistically smaller than those of *RegFls*. This indicates that extreme floods tend to occur concurrently with neighboring basins rather than sporadically over the monsoon region. The concurrency is dictated by the key features of precipitating systems and/or its interactions with regional topography.

(4) Spatial extent and intensity of *RegFls*: *RegFls* are diverse in spatial extent and intensity. The *RegFls* with large spatial extent (i.e., Large *RegFls*) show the largest soil moisture anomalies. There are notable contrasts in the fine-scale structures of

flood-producing storms across different *RegFls* groups, but they are not reflected in basin-average rainfall anomalies. These fine-scale storm structures superimposed on wet soils dictate contrasting flood behaviors over eastern China. This indicates that spatial dependence of rainfall can be translated into flood processes during the monsoon season.

(5) Predicting *RegFls* characteristics: statistical modeling further highlights the importance of antecedent soil moisture and maximum rainfall intensity in dictating *RegFls* severity. While physiographic attributes might play a role in basin-scale





flood responses, it is more critical to capture spatial-temporal patterns of rainfall and soil moisture for large-scale flood modelling and risk analyses.

The core of our analytical framework is *RegFl* identification using a density-based clustering algorithm, with in-situ stream gauging observations as the input. Although our results have been tested by manually modify the density of stream gauging networks over eastern China, it is advisable to apply the algorithm over a stream gauging network with more or less

uniform density. This can be achieved through sampling stations according to basin size or stream order. The hyperparameters need manual adjustments by checking against other established flood archives for the region of interest. A caveat of our study is that only stations with AMFs are used for *RegFl* identification. We use contiguous convex-hull polygon to represent the flood footprint to include neighboring stations that experience smaller floods. We emphasize that our *RegFls* catalog represents the collection of the most severe flood hazards over eastern China.

The proposed framework contributes to flood science by reinforcing the spatial characterization of large-scale flood hazards. This contrasts to conventional flood studies that predominantly rely on derived statistics (e.g., peak discharge, timing, volume) from flood hydrographs at site scales (Blöschl et al., 2017; Blöschl et al., 2019; He et al., 2022). Our framework, with explicitly-defined metrics of flood extent and intensity, provides an alternative approach of modeling large-scale flood hazards and risk assessment. This can be achieved through examining the statistical properties of these flood metrics and its association

with the associated impacts (e.g., economic losses, inundated areas, affected population, etc., see Carozza & Boudreault, 2021, for example).

Our results provide a benchmark dataset for large-scale flood modeling (Del Rio Amador et al., 2023; Carozza and Boudreault, 2021; Gnann et al., 2023). The spatial-temporal clustering pattern of *RegFls* needs to be reproduced before delving into model performance at watershed scales. The ongoing effort includes exploring the link between *RegFls* and large-scale

atmospheric circulations. Upscaling to region-scale processes facilities linking potential flood hazards with synoptic systems rather than dealing with intricate basin-scale flood response. This link can serve as the basis for improved flood risk management (e.g., co-ordination of resources for mitigating and adapting large-scale flood hazards).

**Data and code availability**

The CN05.1 product is available at https://box.nju.edu.cn/f/9e745d4ec4a14d4d94b4/. The ERA5 soil moisture dataset is

obtained from https://cds.climate.copernicus.eu/cdsapp#!/dataset/reanalysis-era5-single-levels/. The IBTrACS dataset is obtained from https://www.ncdc.noaa.gov/ibtracs/. The land use datasets are obtained from the Data Center for Resources and Environmental Sciences, Chinese Academy of Sciences (RESDC-CAS, http://www.resdc.cn/). The digital elevation model is available at https://cgiarcsi.community/data/srtm-90m-digital-elevation-database-v4-1/. The global HydroLAKES dataset, flow direction and accumulation dataset are available at http://www.hydrosheds.org/. The authors declare no conflicts of





interest. The flood dataset used in this study is available at https://doi.org/10.6084/m9.figshare.24636153.v1. All the codes are available at https://doi.org/10.6084/m9.figshare.24637266.v1.

**Author contributions**

YY and LY designed the study and carried out the analysis. YY and LY wrote the manuscript with contribution of JZ and QW. All authors contributed to the discussion.

**Competing interests**

The authors declare that they have no conflict of interest.

**Acknowledgements**

This study is supported by the Natural Science Foundation of China (52379012) and the "GeoX" Interdisciplinary Research Funds for the Frontiers Science Center for Critical Earth Material Cycling, Nanjing University.



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





677  **Appendix**

678  Table A1. List of the top twelve most severe *RegFIs* over eastern China.

| Rank | Date | Location | RFI | Precipitating systems |
|---|---|---|---|---|
| 1 | July 26, 2016 | Central China | 39.43 | Cut-off low, southwestern China vortex and northeastern China vortex |
| 2 | July 19, 1989 | Central and northeastern China | 32.13 | Southwestern China vortex, Typhoon Hope (1989) and northeastern China vortex |
| 3 | August 7, 1996 | Central and northeastern China | 30.84 | Typhoon Herb (1996) and northeastern China vortex |
| 4 | August 1, 2001 | Central and northeastern China | 28.26 | Southwest China vortex, northeast China vortex and Typhoon Toraji (2001), cut-off low |
| 5 | August 3, 2007 | Central and southwestern China | 27.03 | Southwest China vortex and northeast China vortex |
| 6 | August 7, 1992 | Central and northeastern China | 26.35 | Cut-off low and Northeast China vortex |
| 7 | July 15, 1997 | Central and southwestern China | 26.35 | Southwest China vortex and cut-off low |
| 8 | July 14, 1999 | Central and southeastern China | 25.67 | Northeast China vortex |
| 9 | June 25, 2002 | Central and southwestern China | 25.43 | Southwest China vortex and northeast China vortex |
| 10 | September 1, 2003 | Central western China | 24.91 | Cut-off low |
| 11 | July 30, 2012 | Central and northeastern China | 24.90 | Northeast China vortex and Typhoon Damrey (2012) |
| 12 | September 17, 2014 | Central and southwestern China | 24.85 | Typhoon Kalmaegi (2014) |

679





680

Figure A1. Elevation map over eastern China. The black lines show Mt. Taihang and Mt. Qinling. Blue lines show major rivers across China, with name shown on the map.