# Peer review of "Processes and controls of regional floods over eastern China"

_Hydrology and Earth System Sciences, 2024_

## Author Comment (AC1)

**Response to Reviewer #1**

(1) This paper proposed a machine-learning based framework to examine the processes and controls of regional floods over eastern China. Authors utilized the stream station network including observations of annual maximum flood peak during 1980-2017, to analyse flood clusters in spatial extents and intensities. The structure of the paper is clear, however, there are some concerns.

***Response***: Thank you for the overall appreciation of our study. We address all your concerns below on a point-by-point basis. Thanks!

(2) For extreme floods cluster in space and time, it is quite urgent to get the water depth spatial distribution and variation in the short time, like serval days. Instead, authors used a time window-15 days, to analyse the flood frequencies. Could authors demonstrate the extreme flood's distribution in a period and define the water depth for extreme floods?

***Response***: Unfortunately, we do not have water depth data associated with each extreme flood. Definition of regional flood is based on concurrent annual flood peak discharge which typically results in the maximum water depth throughout the year for each station. We use 15-days as the time window to consider river routing processes of larger basins and long-lasting rainfall processes. Using smaller time windows will lead to more regional floods but smaller spatial coverages. We had this discussion in Line 154-156. We do not modify the text. Thanks all the same!

(3) In addition, the AMF denotes annual maximum flood peaks. Authors mentioned "The 15-day time window moves from the first to the last date of AMF occurrences for each year. We thus obtain all qualified clusters" in lines 149-150. Does it mean that only one polygon is selected in every year?

***Response***: The number of polygons selected each year is determined by the timing and location of regional floods. As we have clearly mentioned that there are 8.3 regional flood each year on average (see Line 257). For each regional flood, we use the largest cluster to represent its spatial extent, since small clusters that are overlapped with them are deemed as part of the flood process. We made this clear in the revised manuscript. Thanks!

(4) In the methodology part, it is difficult to understand that why authors choose use the inversed ranks in Equation (1) for AMF to represent the severity of RegFl.

***Response***: We use inversed rank to normalize flood peaks so that flood peaks are comparable among each regional flood. We follow Tarouilly et al. (2021) and do not claim the novelty of this metric. By summarizing the inversed ranks, we can characterize both the spatial extent and intensity of local floods. This is what we defined as the severity of a regional flood. We made this clear in Line 180-187. We hope this makes sense to you. Thanks!

(5) Authors used three machine learning algorithms, DBSCAN, K-means and conditional random

forest for identification, characterization and statistics respectively. For each algorithm, it requires training and test. Could authors show the model performance in each algorithm and discuss the influence of model uncertainty in each step impacting on the following model's training and test?

***Response***: Thank you for this critique! We note that training and test is only needed for conditional random forest (CRF). Based on training and test, the reliability of the established model and attribution can be satisfied. We use out-of-bag error (RMSE and R-squared) to assess the performance of CRF. In CRF, each tree is trained on a bootstrap sample of data. This means that approximately one-third of the observations are left out for each tree. These out-of-bag observations (the samples not included in the bootstrap sample) are used to estimate the prediction error of each tree. These OOB errors are then aggregated to obtain the overall OOB error of the CRF model (Line 222-224).

Both DBSCAN and K-means are unsupervised clustering algorithm. There are evaluation metrics for these algorithms, such as Silhouette score and Davies Bouldin score for K-means. The idea of these algorithms is to classify all samples into different groups rather than justifying the reliability of the algorithm itself (i.e., different from random forest modeling). With this said, we spent great efforts in validating our clustering results (Line 145-148; 157-164; 263-268). For instance, the performance of DBSCAN algorithm is evaluated against existing regional flood database (i.e., DFO) by parameter tuning. Our rationale is to develop a catalog of regional floods over China. The flood catalog serves as the basis for the following empirical analysis that highlights their processes and controls. We hope this addresses your concern. Thanks!

(6) The predictors are in different spatial resolutions and time scales. Could authors provide more details about data preprocess?

***Response***: Thank you for this critique. In terms of time scale, we resample hourly soil moisture into daily scale through summation. We do not preprocess the spatial resolutions of different predictors, since they are directly nudged onto the cascade-union area of each regional flood (i.e., region-average statistics). We made this clear in the revised manuscript (Line 117). Thanks!

**References:**

Tarouilly, E., Li, D., & Lettenmaier, D. P. (2021), Western U.S. superfloods in the recent instrumental record, *Water Resources Research*, *57*(9), e2020WR029287. https://doi.org/10.1029/2020wr029287

---

## Author Comment (AC2)

**Response to Reviewer #2**

(1) This study analyzed the floods in eastern China from a regional perspective. The authors developed a novel approach for identifying the regional floods and attributing these floods to their driving factors. Compared to isolated floods, like flash flooding in a small watershed scale, regional floods often cause more catastrophic disasters. Such studies, like this one, are always encouraged to improve our understanding of characteristics and generating mechanisms of regional (i.e., substantial) floods. However, a minor revision is needed before the manuscript can be recommended for acceptance.

*Response*: Thank you for appreciation of our study. We carefully address all your concerns, and provide a point-by-point response below. Thanks!

(2) For the Identification method, the transition from step 3 (DBSCA clusters; Figure 2c) to step 4 (largest convex-hull polygon within a 15-day window) is very confusing. If multiple clusters have been identified in step 3, will you merge them to a bigger one in step 4? Please revise this part to make it clearer.

*Response*: Small clusters that are overlapped with a big cluster are deemed a single regional flood. This is related to the development and decay of flood process in accompany with rainfall propagation and soil moisture replenishment. We choose the biggest cluster to represent the extent of regional flood, ignoring its evolving process (Line 151-154). We revise the flow chart in Figure 2 to better illustrate our method. Thanks!

(3) The statistical modeling and its associated results are not central to the main content of this manuscript. I suggest removing these elements and considering expanding them into a separate paper.

*Response*: Thank you for this suggestion. We prefer to keep this section in the revised manuscript, since the statistical modeling analyses highlight the importance of different factors in discerning severity of regional floods in a quantitative way. This is a complement to previous empirical attribution analysis. We do not modify the text. Thanks all the same!

(4) In line 181, please briefly explain "flood ratio" as the way how you explain the "unit peak discharge".

*Response*: Done. Thanks!

(5) Consider changing the subtitle 2.3 from "Empirical analyses" to "Regional floods attribution".

*Response*: Done. Thanks!

(6) In section 3.2, please further explain "the role of orographic lifting in enhancing rainfall

intensity." Is there any strong relationship between elevation and rainfall intensity? Also, consider citing the following paper in Section 3.2.

Houze Jr, Robert A. "Orographic effects on precipitating clouds." Reviews of Geophysics 50, no. 1 (2012).

***Response***: The relationship between elevation and rainfall intensity has been detected in previous studies, including the northern China region. This is mainly through enhanced moisture convergence that leads to updraft over the windward region. We add this statement and references (including Houze Jr, 2012) in the revised manuscript (Line 277-281). Thanks!

(7) Consider changing "Mild RegFls" to "Moderate RegFls".

***Response***: Done. Thanks!

(8) In figure 9, it shows that rainfall peaks in about 2 days in advance of soil moisture and peak flow. My interpretation is that extreme rainfall caused the increase in soil moisture. In other words, the watershed antecedent soil moisture does not matter in extreme floods. Please defend it.

***Response***: Thank you for raising this critique. We absolutely agree that the replenishment of soil moisture is mainly through rainfall process. However, the depletion of soil moisture may through evaporation and/or lateral exchange, and is thus less correlated with rainfall. Here we focus on antecedent soil moisture prior to flood peak discharge by following basic hydrological assumption that soil moisture condition regulates runoff-generation processes. Previous studies (e.g., Sharma et al., 2018) also confirm the role of antecedent soil moisture in dominating nonlinear flood response to rainfall changes. We hope our reasoning can address your concern. Thanks!

**References:**

Sharma, A., Wasko, C., & Lettenmaier, D. P. (2018), If precipitation extremes are increasing, why aren't floods?, *Water Resources Research*, *54*(11), 8545-8551. https://doi.org/10.1029/2018wr023749